# Taxonomy and Physiology of *Oxyrrhis marina* and *Oxyrrhis maritima* in Korean Waters

Min Kyoung Jung [1], Tae Yeon Yin [1], Seung Joo Moon [2], Jaeyeon Park [2,*] and Eun Young Yoon [1,*]

1   Climate Change Research Laboratory, Advanced Institute of Convergence Technology, Suwon 16229, Korea; mk_jung@snu.ac.kr (M.K.J.); in_taae@snu.ac.kr (T.Y.Y.)
2   Environment and Resource Convergence Center, Advanced Institute of Convergence Technology, Suwon 16229, Korea; sjmoon04@snu.ac.kr
*   Correspondence: bada0@snu.ac.kr (J.P.); journal04@snu.ac.kr (E.Y.Y.)

**Abstract:** The genus *Oxyrrhis* is a heterotrophic dinoflagellate found in diverse marine environments. *Oxyrrhis* spp. have received attention owing to their ecological and industrial importance, high lipid contents, and docosahexaenoic acid formation. To the best of our knowledge, contrary to *O. marina*, ecophysiological characterization studies on *O. maritima* have not yet been reported. Therefore, we investigated the taxonomy and ecophysiology of four strains of *O. marina* from coastal waters and two strains of *O. maritima* from the littoral tidepool waters of Korea. Based on phylogenetic trees constructed using internal transcribed spacer ribosomal DNA (ITS rDNA) and SSU rDNA of dinoflagellates, the clade of all four *O. marina* strains was divergent from that of the two *O. maritima* strains. We measured the growth rates of both species at various water temperatures (10–36 °C), salinities (5–90), and light intensities (0–100 μE·m$^{-2}$·s$^{-1}$). The lowest (*O. marina* and *O. maritima*: 10 °C) and highest temperatures (*O. marina*: <35 °C, *O. maritima*: >35 °C) revealed that *O. maritima* has more tolerance to high salinity. This study provides a basis for understanding the ecophysiology of *O. marina* and *O. maritima* and their population dynamics in marine ecosystems.

**Keywords:** heterotrophic dinoflagellates; Korean waters; *Oxyrrhis*; physiology; taxonomy

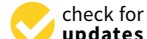



## 1. Introduction

Heterotrophic dinoflagellates (HTDs) are of great importance in the marine food web because they coexist as predators and prey for other species [1]. In this study, we focused on HTDs *Oxyrrhis marina* (*O. marina*) and *Oxyrrhis maritima* (*O. maritima*). There are advantages to using *Oxyrrhis* as a model for other HTD natural samples, to identify and manipulate in experiments, and hence, to study the model of evolution and exploitation of this species. *Oxyrrhis* is a widespread, free-living, and ecologically important HTD [2–5] that has been utilized as an important model organism for a broad range of studies [6–8].

The genus *Oxyrrhis* is commonly found in marine and brackish nearshore waters, tide pools, and salterns. However, it explicitly excluded dinokaryotes in common dinoflagellate classification schemes [9]. Unlike most dinoflagellates, which either have a longitudinal and a transverse flagellum emerging from the sulcus and the cingulum, respectively, or both flagella growing from the apical area (Prorocentrales) [4], *Oxyrrhis* have both flagella emerging from the ventral side. Therefore, the phylogenetic position of *O. marina* remains controversial. While some of the morphological and cytological studies support its basal position in the dinoflagellate classification of the higher ranks [10], others infer a highly derived position within the order of Gonyaulacales, and the majority of recent studies support *O. marina* as an ancestral dinoflagellate [11].

Morphological studies have raised disputes on whether the genus *Oxyrrhis* contains multiple species (*O. marina*, *O. maritima* van Meel 1969, *O. phaeocysticola* Scherffel, 1900, and *O. tentaculifera* Conrad, 1939) or only one species (*O. marina*) [4,12–17]. In the literature, the lengths of *O. marina* Dujardin (1841) and *O. marina* Kent (1880) are 28–51 μm, whereas

that of *O. maritima* Van Meel (1969) is 16–24 μm (Table 1). Recently, molecular phylogenetic studies have favored the notion that two sibling species exist in this genus. Lowe et al. suggested that this genus consists of two species, *O. marina* and *O. maritima*, and each species consists of two clades [4].

**Table 1.** The designations, and taxonomic authorities, for species in the genus *Oxyrrhis*.

| *Oxyrrhis* | Year | Length (μm) | Flagella | Shape | Location |
|---|---|---|---|---|---|
| *O. marina* Dujardin | 1841 | 44 | Several | Oblong, oval bodied, rounded posteriorly | Mediterranean |
| *O. marina* Kent | 1880 | 28–51 | 2 | Body conical, sub-cylindrical, rounded posteriorly | St Helier, Jersey |
| *O. marina* (starved) | 2001 | 16–31 | 2 | Oblong, oval bodied, rounded posteriorly | Gunsan, Korea |
| *O. phaeocysticola* | 1900 | 20 | 2 | Rounded posterior, pointed anterior, excavated oral region with trunk-like projection | Helgoland, Germany |
| *O. tentaculifera* Conrad | 1939 | 38 | 2 | More voluminous than *O. marina* | Belgium |
| *O. maritima* Van Meel | 1969 | 16–24 | 2 | Twice as long as wide, has a tentacle | Belgium |
| *O. maritima* (starved) | 2013 | 15–28 | 2 | Body conical, sub-cylindrical, rounded posteriorly | Jeju, Korea |

However, the existing data are scattered, despite the increasing number of studies on *O. marina* [11]. A myriad of ecological, physiological, and genetic studies have been conducted without accurate identification of the species. Thus, we established four strains of *O. marina* and two strains of *O. maritima* by isolating and culturing from six different locations in the Korean environmental water samples. The morphological features, phylogeny, and nuclear ribosomal DNA were analyzed based on the six strains of *Oxyrrhis* spp.

The growth rate of heterotrophic protists is affected by various abiotic factors, such as salinity [18,19], temperature [20], and light intensity [21]. Thus, the physiological study of each species is a critical step in understanding the structure and function of the ocean ecosystem. There have been many studies on the effects of temperature, salinity, and light conditions on the growth rates of *O. marina* [22–24]. However, there have been no studies on the ecophysiological characterizations of *O. maritima*. Therefore, it is worthwhile to explore this topic. Moreover, the difference between the ecophysiological characterizations of these two *Oxyrrhis* spp. is poorly understood. Thus, we measured and compared the growth rates of *O. marina* and *O. maritima*.

In the present study, we measured the growth rates of *O. marina* and *O. maritima* at various water temperatures (10–36 °C), salinities (5–90), and light intensities (0–100 μE·m$^{-2}$·s$^{-1}$). In particular, we tested whether *O. marina* could grow in darkness when suitable prey was provided. Furthermore, we determined the lowest and highest temperatures and salinities at which *O. marina* and *O. maritima* can survive or grow. The results of the present study provide a basis for understanding the ecophysiology of *O. marina* and *O. maritima* and their population dynamics in marine ecosystems.

## 2. Materials and Methods

### 2.1. Collection, Isolation, and Culturing of Oxyrrhis spp.

Water samples were collected using a syringe tubing and water sampling bucket from the surface of six different stations at littoral tide pools in Jeju Island and coastal waters in Korea (Shiwha, Gunsan, Masan, and Karorim; Table 2; Figure 1) between May 2001 and May 2013. Water temperature and salinity measured during sampling for *O. marina* were 13.5–19.7 °C and 27.7–33 °C, respectively, but no physical information was available in the case of *O. maritima*.

**Table 2.** Environmental conditions of collected *Oxyrrhis* spp. in the coastal waters of Korea.

| Location | Date | GPS | Temperature (°C) | Salinity | Habitats | Strain Name |
|---|---|---|---|---|---|---|
| Shiwha (SH) | November 2011 | 37°18′00.1″ N 126°40′19.5″ E | 19.7 | 31.0 | Coastal | OMSH_1111 |
| Karorim (KRR) | May 2010 | 36°45′19.1″ N 126°11′28.0″ E | 19.5 | 33.0 | Saltern | OMKRR_1005 |
| Gunsan (GS) | May 2001 | 35°56′20.1″ N 126°31′50.5″ E | 16.0 | 27.7 | Coastal | OMGS_0105 |
| Masan (MS) | April 2010 | 35°11′41.2″ N 128°34′23.3″ E | 13.5 | 29.8 | Coastal | OMMS_1004 |
| Jeju Island (JJ) | May 2013 | 33°12′47.7″ N 126°17′40.6″ E | - | - | Tide pool | OMJJ_1305 |
| Jeju Island (HD) | May 2013 | 33°32′33.6″ N 126°40′10.7″ E | - | - | Tide pool | OMHD_1305 |

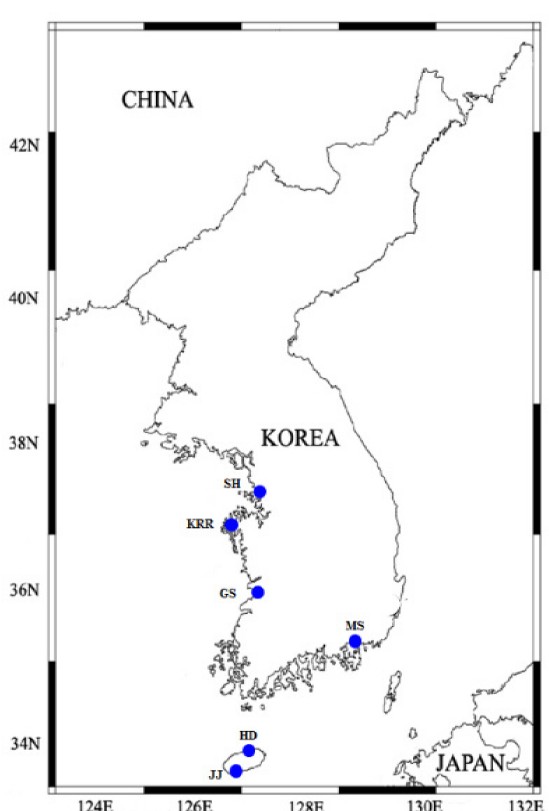

**Figure 1.** Map of the sampling stations of *Oxyrrhis marina* (Shiwha [SH], Karorim [KRR], Gunsan [GS], Masan [MS], Korea) and *O. maritima* (Jeju-Anduck [JJ] and Hamduck [HD], which are located in Jeju Island, Korea).

The sample was filtered through a 154-μm Nitex mesh screen and placed in six-well tissue culture plates. A clonal culture was established following two serial single-cell isolations, and the dried yeast (*Saccharomyces cerevisiae*) obtained from Red Star (Lesaffre Yeast Corporation, Milwaukee, WI, USA) were provided as food. As the culture volume increased, cells were transferred to 32-, 250-, and 500-mL polycarbonate (PC) bottles. The bottles were placed on a shelf at 20 °C under continuous illumination of 10 μE·m$^{-2}$·s$^{-1}$ provided by a cool white fluorescent light source. As the cultures became dense, approximately every 2–3 d, they were transferred to new 500-mL PC bottles containing dried yeast as prey.

### 2.2. Microscopic Observations

The morphologies of *O. marina* and *O. maritima* were examined using light microscopy. The length and width of live vegetative flagellated cells were measured using an image analysis system for processing images taken with a compound microscope (BX53 microscope, Olympus, Japan).

### 2.3. DNA Extraction, PCR Amplification, and Sequencing

Three to five cells of *O. marina* and *O. maritima* from a clonal culture were transferred to a 0.2-mL PCR tube containing 38.75 μL distilled water. To lyse the cell membranes before PCR analysis, the tube was frozen at −72 °C for 1–3 min and then thawed. The final mixture (50 μL) was vortexed, and PCR was performed in a thermal cycler (Eppendorf AG, Mastercycler® ep, model 5341, Hamburg, Germany). The final concentrations of PCR products were: 5 μL of 10X F-Star Taq buffer, 1 μL of 10 mM of dNTP mix, 0.25 μL of 5 U/μL BioFACT™ F-Star Taq DNA polymerase (BioFACT Co., Ltd., Daejeon, Korea), and 0.02 μM of each primer (Table S1) and a final volume of 50 μL was made. The PCR conditions were as follows: one activation step at 95 °C for 2 min, followed by 40 cycles at 95 °C for 20 s, the selected annealing temperature for 40 s, and 72 °C for 1 min, and a final elongation step at 72 °C for 5 min. The annealing temperature was adjusted depending on the primers used according to the manufacturer's instructions. Positive and negative controls were used for all the amplification reactions. The purity of the amplicons was checked by performing electrophoresis of 5 μL of PCR products mixed with 1 μL of red fluorescent reagent (Redstar, Daejeon, Korea) using a 0.9% agarose gel at 80 V and observing it under a UV lamp to ensure that a single product was formed. Products containing a single band were then purified using an AccuPrep® PCR purification kit (Bioneer Corp., Daejeon, Korea) according to the manufacturer's instructions. The purified amplicon was sent to Biomedic Corporation (Gyeonggi, Korea) for sequencing using an ABI PRISM® 3700 DNA Analyzer (Applied Biosystems, Foster City, CA, USA). Each portion of the target DNA was independently sequenced three times in both directions using primer pairs identical to those used for DNA amplification. These sequences were aligned using the ContigExpress alignment program (InforMax, Frederick, MD, USA) to remove low-quality regions and to assemble the individual sequence reads.

### 2.4. Phylogenetic Analysis

Phylogenetic analyses were performed based on alignments of the partial 3-end small subunit (SSU) and internal transcribed spacer region (ITS1-5.8S-ITS2) sequences. All sequence regions are located in the nuclear genome. Analyses included sequences obtained from GenBank and the present study (Table S2). Multiple sequences were aligned automatically using MEGA-X native implementation of ClustalW [25], and further aligned manually to refine the alignments. Two data sets were analyzed as follows: (1) SSU sequences only, (2) ITS1-5.8S-ITS2 sequences only. SSU sequences were used to investigate the systematic distance between species. Meanwhile, ITS1-5.8S-ITS2 sequences were used to determine intraspecific genetic diversity between *Oxyrrhis* species. The final alignment of the SSU rDNA contained 19 dinoflagellate species, plus *Perkinsus*, *Apicomplexans*, Ciliate, and *Stramenopiles* sequences that were used as outgroups. The final alignment of the ITS1-

5.8S-ITS2 rDNA contained 33 *Oxyrrhis* sequences, plus *Perkinsus*, *Apicomplexans*, Ciliate, and *Stramenopiles* sequences that were used as outgroups. Two main model-based methods were used to infer phylogenetic relationships: maximum likelihood (ML) and Bayesian inference based on the likelihood function. ML analysis was performed with the RAxML 7.0.3 program [26] using the default GTR + G model. Tree likelihoods were estimated using a heuristic search with 200 random additional sequence replicates and tree bisection and reconnection branch swapping. ML bootstrapping with 1000 replications was also conducted. Bayesian analysis was performed with MrBayes v.3.1.2 [27,28] using the GTR + G + I model according to the model test results. The program was set to operate with a gamma distribution (+G) with five rate categories and by assuming that a certain fraction of sites is evolutionarily invariable(+I), and four Monte Carlo Markov chains (MCMC) starting from a random tree. In total, 2,000,000 generations were calculated with trees sampled every 100 generations and with a prior burn-in of 5000 generations.

### 2.5. Physiology

Experiments 1 and 2 were designed to investigate the effects of temperatures on the growth rates of *O. marina* (GS) and *O. maritima* (JJ) at 5, 10, 15, 20, 25, 30, 35, and 36 °C.

Each of the 5 L PC bottles containing *O. marina* or *O. maritima* was maintained at the target temperature and 20 $\mu E \cdot m^{-2} \cdot s^{-1}$ of continuous cool white fluorescent continuous light. *O. marina* and *O. maritima* cells were acclimated to the target temperatures, except <5 °C and >35 °C, for more than 10 d. For the target temperatures of <5 °C and >35 °C, cells acclimated at 10 °C and 30 °C for 10 d, respectively, were used without further acclimation. Dried yeast was provided as prey every 1–2 d. During this pre-incubation period, one-mL aliquots from each bottle were taken every day to determine the abundance of *O. marina* or *O. maritima* using a compound microscope.

In experiments 1 and 2, the initial abundances of *O. marina* and *O. maritima* (ca. 200 cell mL$^{-1}$ at temperatures of 15, 20, 25, and 30 °C; 700 cells mL$^{-1}$ at 10, 32, 35, and 36 °C) and the concentrations of autoclaved dried yeasts (ca. 20 mg per liter) were established using an auto-pipette to deliver predetermined volumes of known cell concentrations to the bottles.

Experiments 3 and 4 were designed to investigate the effects of salinity on the growth rates of *O. marina* and *O. maritima*. For the experiments, artificial seawater (S 9883 Sea salts, Sigma-Aldrich, St. Louis, MO, USA) or sterilized distilled water was added to sterilized seawater to obtain final salinities of 5, 10, 20, 33, 50, 70, and 90. Each bottle was placed on a shelf in a culture room maintained at 20 °C. The bottles were fed with dried yeasts as prey every 1–2 d. *O. marina* and *O. maritima* cells were acclimated to the target salinities, except <5 and >70, for more than 10 d. For target salinities <5 and >70, the cells acclimated at a salinity of 10 and 70 for 10 d, respectively, were used without further acclimation.

In experiments 3 and 4, the initial abundances of *O. marina* and *O. maritima* (ca. 300 cell mL$^{-1}$ at salinities of 10, 20, 33, and 50; 500 cell mL$^{-1}$ at salinities of 10, 70, and 90) and the concentrations of autoclaved dried yeast (ca. 20 mg dried yeast per liter) were established using an auto-pipette to deliver predetermined volumes of known cell concentrations to the bottles.

Experiments 5 and 6 were conducted to investigate the effects of light intensity on the growth rates of *O. marina* and *O. maritima* when fed with dried yeast (ca. 4 mg dried yeast per 200 mL) as prey. Dense cultures of *O. marina* and *O. maritima* were grown on dried yeast at 10 $\mu E \cdot m^{-2} \cdot s^{-1}$ of continuous cool white fluorescent light without any mixing. Bottles were wrapped with aluminum foil for darkness and placed on the shelf for each target light intensity of 0, 1, 20, or 100 $\mu E \cdot m^{-2} \cdot s^{-1}$. The experimental bottles were maintained at 20 °C, and every 1–2 d, the contents of each bottle were evenly distributed into two 1-L PC bottles containing the target amount of dried yeast and acclimated to their target light intensities for 10 d.

To determine the actual initial predator densities (cells mL$^{-1}$) at the beginning of the experiment and after incubation, 5 mL aliquots were removed from each bottle and fixed

with 5% Lugol's solution, and all *O. marina* or *O. maritima* cells and (or >300) prey cells from the triplicate SRCs were enumerated. Before taking subsamples, the conditions of *O. marina*, *O. maritima*, and prey were assessed under a dissecting microscope.

The specific growth rate of *O. marina* or *O. maritima*, μ (d$^{-1}$), was calculated as follows:

$$\mu = \ln (H_t/H_0)/t, \tag{1}$$

where $H_0$ is the initial concentration (cells mL$^{-1}$) of *O. marina* or *O. maritima*, and $H_t$ is the final concentration after time t (d$^{-1}$).

## 3. Results

### 3.1. Morphology

#### 3.1.1. Morphology of *O. marina*

Under a light microscope (LM), the living cell was subovoid, asymmetrical posteriorly; girdle postmedial, not extending to the dorsal surface; sulcus spreading posteroventrally; flagella midventral; tentacular lobe situated between two flagella, dividing the broad undeveloped ventral sulcus (Figure 2). It appeared colorless, but had pink pigmentation, which was apparent in concentrated cultures. Therefore, this species is called a "pink pig" because of the pink colored water at a high concentration of the organisms [29].

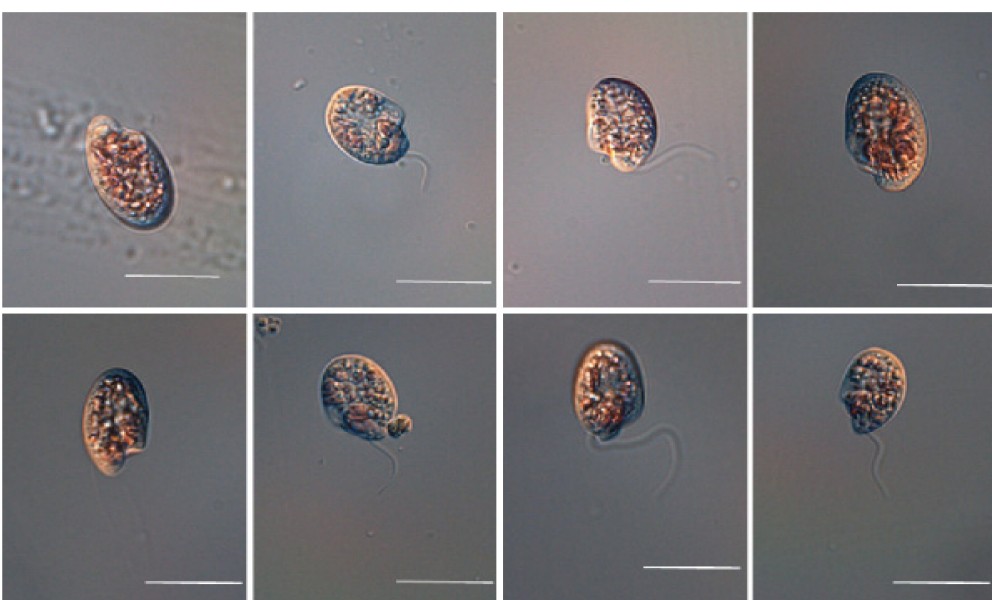

**Figure 2.** Micrographs of *Oxyrrhis marina* (OMGS 0105) using light microscopy. Scale bars = 20 μm.

The length and width of living cells of starved *O. marina* (n = 30) ranged 16.0–31.2 μm (mean ± SD = 23.6 ± 4.2 μm) and 8.0–19.2 μm (mean ± SD = 13.8 ± 2.4 μm), respectively, whereas live cells fed with *Prorocentrum minimum* (n = 15) measured 15.2–33.7 μm (28.4 ± 4.64 μm) and 15.6–24.0 μm (18.9 ± 3.7 μm) in length and width, respectively. Thus, the ratio of the cell length to width of living cells starved for 5 d (mean ± SD = 1.74 ± 0.23; range = 1.2–2.2; n = 30) was larger than that of cells fed with *P. minimum* (mean ± SD = 1.53 ± 0.17; range = 1.3–1.8; n = 15).

The mean equivalent spherical diameter (ESD, mean ± SD) was measured using an electronic particle counter (Coulter Multisizer II; Coulter Corp., Miami, FL, USA). The ESD of *O. marina* (OMGS 0105) starved for 5d (n > 2000) was 17.11–22.72 μm (19.85 ± 2.54 μm).

#### 3.1.2. Morphology of *O. maritima*

As shown in Figure 3, under LM, living cells were observed to be subovoid, posteriorly asymmetrical; girdle postmedial, not extending to the dorsal surface; sulcus spreading

posteroventrally; flagella midventral; tentacular lobe present between two flagella, dividing the broad undeveloped ventral sulcus. It appeared colorless but had pink pigmentation, which was apparent in concentrated cultures.

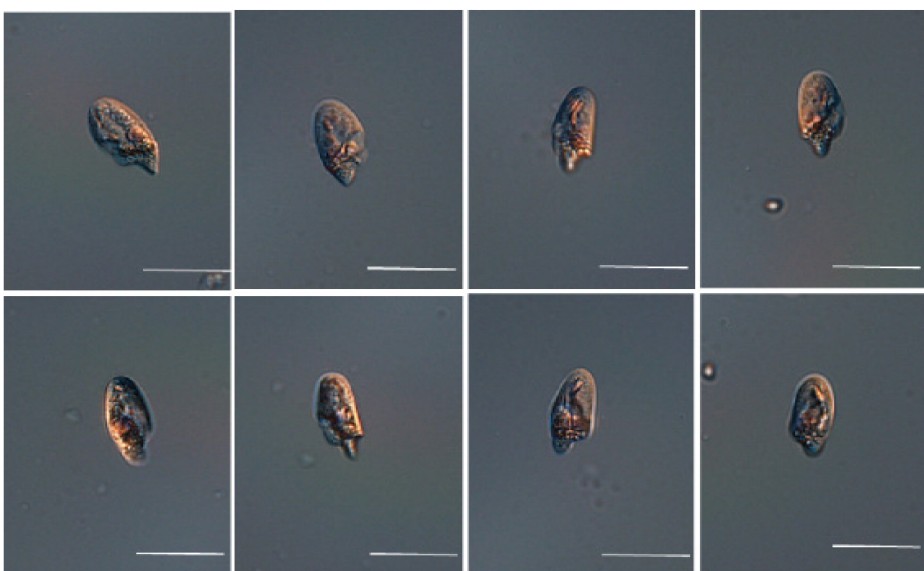

**Figure 3.** Micrographs of *Oxyrrhis maritima* (OMJJ 305) using light microscopy. Scale bars = 20 μm.

The length and width of living cells of *O. maritima* starved for 5 d (n = 30) ranged 14.4–28.0 μm (mean + SD = 20.6 ± 3.3 μm) and 9.6–15.2 μm (11.8 ± 1.9 μm), respectively, whereas live cells fed with *P. minimum* (n = 15) measured 16.8–35.2 μm (27.6 ± 5.1 μm) and 11.2–24.7 μm (19.9 ± 3.6 μm) in length and width, respectively. Thus, the ratio of length to width of living cells starved for 5 d (mean ± SD = 1.74 ± 0.23; range = 1.2–2.2; n = 30) was larger than that of the cells of *O. maritima* fed with *P. minimum* (mean ± SD = 1.53 ± 0.17; range = 1.3–1.8; n = 15). The ESD of *O. maritima* (OMJJ 1305) measured 11.4–15.1 μm (13.3 ± 1.5 μm), respectively. The ESD of *O. maritima* was approximately 45% smaller than that of *O. marina*.

### 3.2. DNA Sequence Analysis (Genetic Variability)

Nuclear sequences of SSU, ITS, and LSU rDNA were obtained from four strains of *O. marina* and two strains of *O. maritima*. The SSU, ITS 1 and 2, 5.8S, and LSU rDNA sequences of six strains of *Oxyrrhis* spp. from Gunsan (GS), Karorim (KRR), Masan (MS), Shiwha (SH), Jeju Anduck (JJ), and Jeju Hamduck (HD) comprised of 2882; 3010; 3033; 3006; 3051; and 3038 nucleotides, respectively.

Sequence alignments were performed using the MEGA-X native implementation of ClustalW. Sequence divergence was exceptionally high among some strains (Table 3). We compared the genetic divergence (%), except gaps, of the total sequences of the six strains of *Oxyrrhis* spp.

When aligned the sequence using the NCBI blast, the total rDNA sequence of *O. maritima* (JJ) differed greatly from those of *O. marina* isolated from KRR, SH, MS, and GS by 445, 494, 433, and 535 bp (except Gap), and a similarity of 82.2%, 81.2%, 82.1%, and 77.5%, respectively (corresponding to 17.8–22.5% dissimilarity).

The difference among the sequences of the four strains of *O. marina* was 0.1–0.3%, whereas that between the two strains of *O. maritima* was only 0.1%. However, the difference between the sequences of *O. marina* and *O. maritima* was 17.8–22.5%. In the phylogenetic trees based on the ITS1-5.8s-ITS2 rDNA of dinoflagellates, the clade of all four strains of *O. marina* belonged to lineage I (clades 1 and 2) and was clearly divergent from that of the two strains of *O. maritima* that belonged to lineage II (clade 4).

**Table 3.** Comparison of the total sequences of *O. marina* (KRR, SH, MS, GS) and *O. maritima* (JJ and HD). The numbers are base pairs different from each other. The numbers in parenthesis are similar (%) except gaps.

| Species | *O. marina* | | | | *O. maritima* | |
|---|---|---|---|---|---|---|
| Location | KRR | SH | MS | GS | JJ | HD |
| KRR | 0 (100%) | | | | | |
| SH | 215 (92.9%) | 0 (100%) | | | | |
| MS | 202 (93.3%) | 2 (99.9%) | 0 (100%) | | | |
| GS | 4 (99.9%) | 211 (92.7%) | 201 (93.1%) | 0 (100%) | | |
| JJ | 445 (82.2%) | 494 (81.2%) | 433 (82.1%) | 535 (77.5%) | 0 (100%) | |
| HD | 427 (83.0%) | 492 (81.3%) | 430 (82.3%) | 421 (82.3%) | 4 (99.9%) | 0 (100%) |

*3.3. Phylogeny*

Figure 4 shows the phylogenetic trees constructed based on SSU rDNA sequences from nucleotide DNA to identify the phylogenetic position of *Oxyrrhis* spp. This confirmed that *Oxyrrhis* spp. were extremely divergent from core dinoflagellates, such as *Alexandrium* spp., and *Amphidinium* spp. (Figure 4). *Oxyrrhis* branched after *Perkinsus marinus*, which is between the dinoflagellates and Apicomplexans, with high bootstrap support in Bayesian analysis. Moreover, its branches proved that the *Oxyrrhis* was positioned at the base of dinoflagellate lineages. The branch length of SSU rDNA sequences of *Oxyrrhis* was approximately eight times as long as that of *P. marinus*. In all phylogenetic analyses, *Oxyrrhis* had two highly different species, and the four strains of *O. marina* and two strains of *O. maritima* branched together with high support. We used only *Stramenopiles* as outgroups, but Apicomplexans and Ciliates were apparently divergent with over 0.99 (Bayesian) probability.

Furthermore, the phylogenetic tree based on ITS rDNA sequences proved the two highly divergent lineages within *O. marina* morphospecies (Figure 5). The two *Oxyrrhis* lineages divided four clades with high bootstrap support as follows: lineages I and II—1.00 (Bayesian)/100 (ML), clades 1 and 2—0.92 (Bayesian)/75 (ML), clades 3 and 4—0.99 (Bayesian)/77 (ML). According to the pairwise distance analysis, the net distance between clades 1 and 2 was 0.219, and this was 0.309 between clades 3 and 4. Moreover, the longest distance between clades was 0.456 between clades 1 and 3. Clades 1, 2, and 3 had a very short distance within each clade, but clade 4 had a slightly longer distance (0.32) than the others. Based on Lowe et al., clades 1 and 2 were classified as *O. marina* and clades 3 and 4 was *O. maritima*. The JJ strain formed a clade within the *O. maritima* HD, and both strains were closely related to *O. maritima* KC012357 and *O. marina* CCAP 1133/4, which were assigned to clade 4 in Lowe et al. In addition, other *O. maritima* species belonged to clade 3 together with two *O. marina* strains (FJ853668, FJ853677), and *O. marina* strains, except for these two strains, belonged to clades 1 and 2, respectively. In addition, four strains isolated from Korean coastal waters showed a closer relationship with each other. Thus, these two species are genetically different from each other, and the strains of *Oxyrrhis* in the Korean coastal waters are split into three clades.

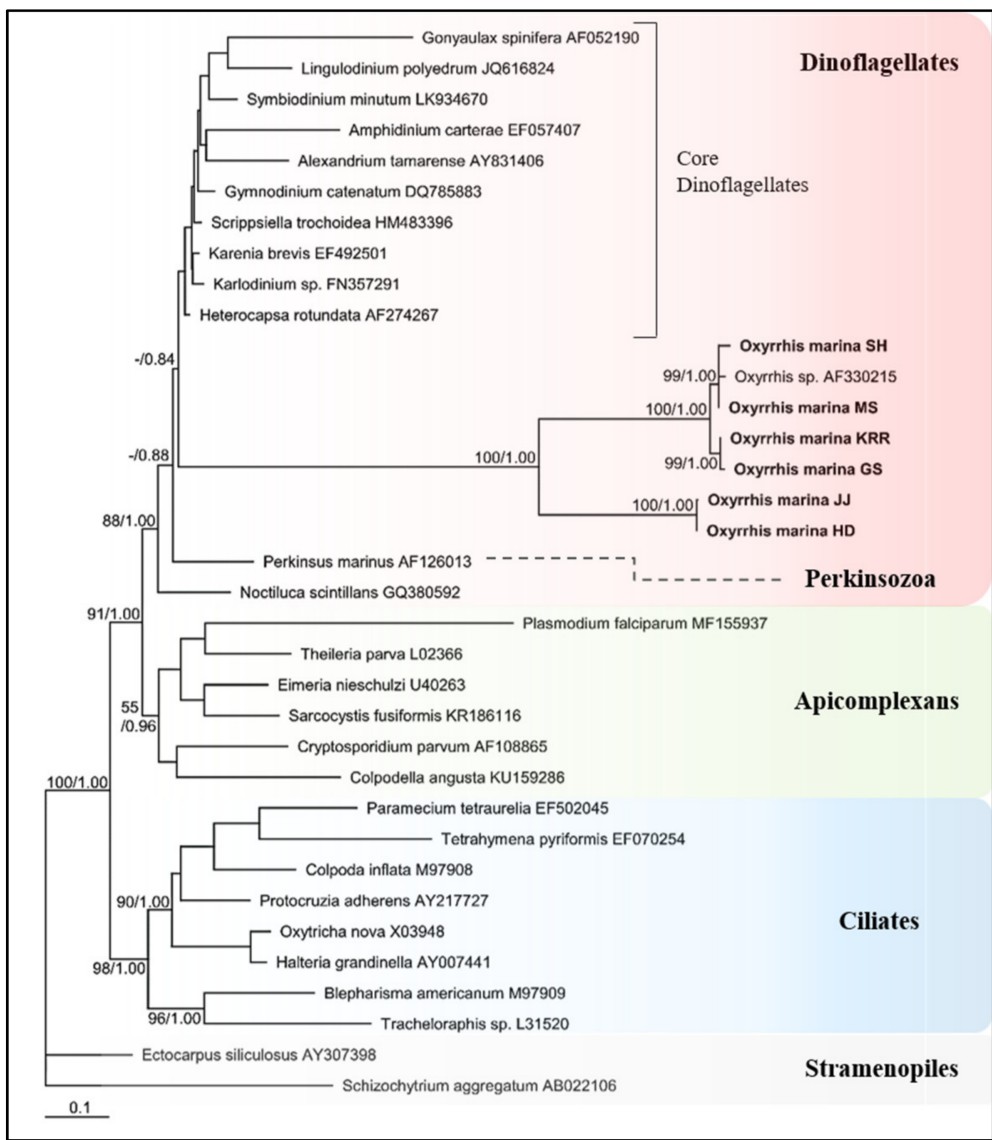

**Figure 4.** Consensus Maximum likelihood tree of nuclear small subunit (SSU) sequences for 6 strains of *Oxyrrhis* spp. with other Dinoflagellates, Perkinsozoa and Apicomplexans, Ciliates and outgroup Stramenopiles. The branch lengths are proportional to the number of base changes. The numbers near the branches indicate the Bayesian posterior probability (**left**) and Maximum likelihood bootstrap values (**right**). Posterior probabilities ≥0.5 are shown.

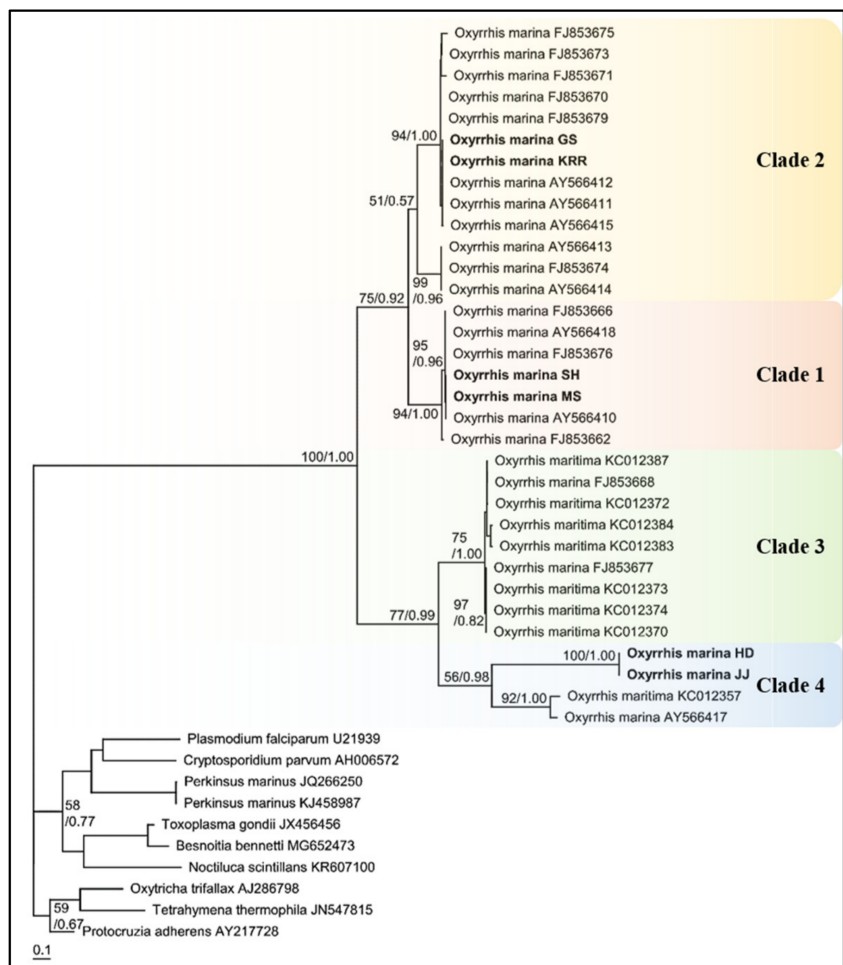

**Figure 5.** Consensus Maximum likelihood tree of nuclear internal transcribed spacer 1 (ITS1)-5.8S-ITS2 sequences for 6 strains of *Oxyrrhis* spp. with other dinoflagellates and outgroup, Perkinsus sp., Ciliates, and Apicomplexans. The branch lengths are proportional to the number of base changes. The numbers near the branches indicate the Bayesian posterior probability (**left**) and maximum likelihood bootstrap values (**right**). Posterior probabilities ≥0.5 are shown.

*3.4. Physiology*

*O. marina* did not grow at temperatures >35 °C (Figure 6a). With increasing temperature from 10 to 25 °C, the growth rate of *O. marina* increased continuously from 0.26 d$^{-1}$ at 10 °C to 0.96 d$^{-1}$ at 25 °C, but it rapidly decreased between 30 and 35 °C (Figure 6a). However, *O. maritima* strains grew at temperatures >35 °C. Furthermore, the optimal temperature for supporting the maximum growth rate of *O. marina* was 25 °C, whereas that of *O. maritima* was 30 °C. With increasing temperature from 10 to 30 °C, the growth rate of *O. maritima* increased continuously from 0.22 d$^{-1}$ at 10 °C to 1.11 d$^{-1}$ at 30 °C, but it rapidly decreased between 32 and 36 °C. The growth rates of *O. marina* and *O. maritima* were significantly affected by temperature (ANOVA test, $p < 0.05$, Figure 6a). At 35 °C, the growth rate of *O. maritima* was significantly greater than that of *O. marina* ($p < 0.05$, one-tailed t-test). However, at temperatures of 10, 15, 20, 25, 30, and 32 °C, the growth rates of *O. marina* were not significantly different from those of *O. maritima* ($p > 0.05$, two-tailed t-test).

*O. marina* did not grow at salinities <4. With increasing salinity from 5–50, the growth rate of *O. marina* increased from 0.37 d$^{-1}$ at 5 to 0.67 d$^{-1}$ at 50, but it rapidly decreased between 50 and 90 (Figure 6b). However, *O. maritima* could grow at a salinity of 2. The growth rates of *O. marina* and *O. maritima* were significantly affected by salinity (ANOVA test, $p < 0.05$, Figure 6b). At salinities of 5, 10, and 20, the growth rates of *O. maritima* were

significantly greater than those of *O. marina* ($p < 0.05$, one-tailed t-test), whereas at a salinity of 90, the growth rate of *O. maritima* was significantly lower than that of *O. marina* ($p < 0.05$, one-tailed *t*-test). However, at salinities of 33, 50, and 70, the growth rates of *O. marina* were not significantly different from those of *O. maritima* ($p > 0.05$, two-tailed *t*-test).

At the light intensities of 0–100 $\mu E \cdot m^{-2} \cdot s^{-1}$, the growth rates of dried yeast-fed *O. marina* did not largely change (i.e., 0.58 $d^{-1}$ at 0 $\mu E \cdot m^{-2} \cdot s^{-1}$ to 0.59 $d^{-1}$ at 100 $\mu E \cdot m^{-2} \cdot s^{-1}$). However, with increasing light intensity from 0 to 100 $\mu E \cdot m^{-2} \cdot s^{-1}$, the growth rate of *O. maritima* decreased from 0.85 $d^{-1}$ at 0 $\mu E \cdot m^{-2} \cdot s^{-1}$ to 0.59 $d^{-1}$ at 100 $\mu E \cdot m^{-2} \cdot s^{-1}$ (Figure 6c). The growth rates of *O. marina* and *O. maritima* were not affected by the light intensity (ANOVA test, $p > 0.1$, Figure 6c). In addition, the growth rate of *O. marina* was not significantly different from that of *O. maritima* ($p > 0.1$, two-tailed t-test).

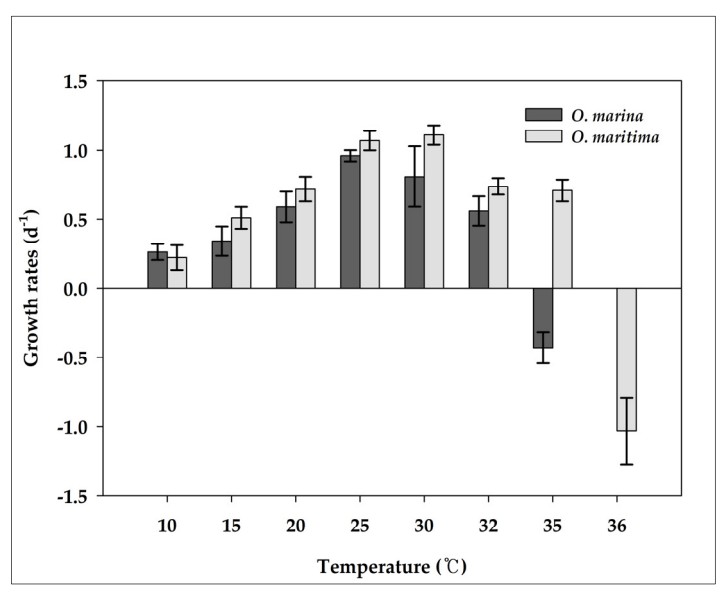

(**a**)

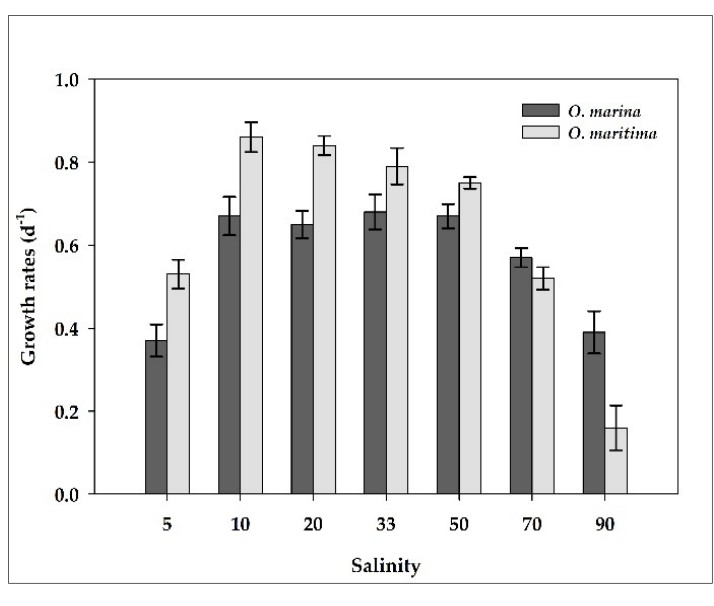

(**b**)

**Figure 6.** *Cont.*

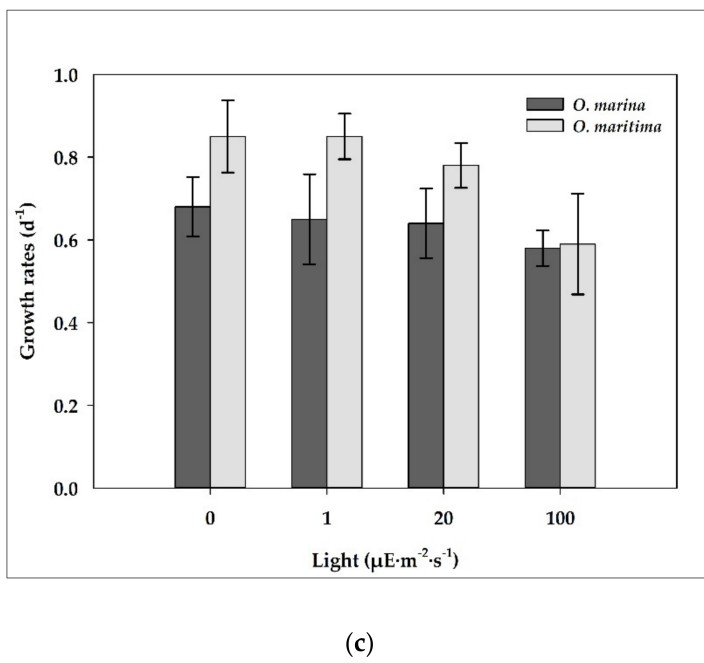

(**c**)

**Figure 6.** The growth rates $(d^{-1})$ of *O. marina* (dark gray) and *O. maritime* (light gray) under various conditions; (**a**) Temperature; (**b**) Salinity; (**c**) Light. The error bar represents the standard deviation from the mean of triplicate data (n = 3).

## 4. Discussion

### 4.1. Comparison of Cell Size

In the literature, the lengths of *O. marina* Dujardin (1841) and *O. marina* Kent (1880) are 28–51 μm, whereas those of *O. maritima* Van Meel (1969) are 16–24 μm (Table 1). Some studies have reported that *O. maritima* could be distinguished from *O. marina* by its rounder and larger shape [4]. However, in contrast to previous studies, the species collected in this study showed that *O. maritima* was longer and sharper than *O. marina*. In the present study, the lengths and ESD of *O. maritima* were approximately 14% and 45% smaller than that of *O. marina*, respectively. Most of the studies have analyzed the genetic diversity of *Oxyrrhis* strains, either by providing only one genetic or morphological information about *O. maritima*, or based on data from strains collected long ago. The results of this study confirmed that the cell size of *O. maritima* is generally smaller than that of *O. marina* by direct analysis of the species collected within 10 years. Additionally, this study attempted to construct integrated data for species classification of *Oxyrrhis* strains by linking morphological and genetic information.

### 4.2. Phylogeny

The phylogenetic positions of *O. marina* are great of importance for understanding cell biology and evolutionary diversity. In particular, the relationships between the Apicomplexa and dinoflagellates have a great impact on disease-causing Apicomplexans and fundamental diversity in eukaryotic cell biology [30]. The dinoflagellate lineage includes abstruse organisms such as syndinians, noctilucoids, Pavilucifera, Perkinsus, and Oxyrrhis. These protists are highly related to Oxyrrhis and their phylogenetic positions are still a matter of debate [31]. Thus, our study conducted the phylogenetic analysis of Oxyrrhis using nuclear SSU rDNA with P. marinus, Noctiluca scintillans, the Apicomplexa (Colphodella), and Ciliates. The relationship with Amoebophrya spp. (Syndinians) was also considered (not shown data), but it was excluded since the sequence length was too long to align with others.

In this study, the phylogenetic tree based on nuclear SSU rDNA sequences was divided into Apicomplexa and dinoflagellates, and then branches of *Noctiluca*, *Perkinsus*, *Oxyrrhis*,

and other dinoflagellates. The internal node, where *Oxyrrhis* and other core dinoflagellates were bound together, proved that *Oxyrrhis* occupies a basal position in the dinoflagellate lineage. Moreover, branches classified as *Oxyrrhis* JJ with HD, SH with MS, and GS with KRR prove that there is a big difference for each clade in the SSU sequence as well.

However, the bootstrapping number was low and the branch length of *Oxyrrhis* was irregularly long. This result is from the limit of the nuclear SSU rDNA. Ribosomal DNA has been utilized in myriad of studies since its advantage is the number of taxa or alignment depth, but it has limitations in that the alignment length is limited, and calculations become more difficult with insertions and deletions in the rDNA sequence. Moreover, some studies asserted that *O. marina* and *N. scintillans* can invalidate phylogeny because of their long branches [32,33]. To solve these limitations and determine the more accurate phylogenetic location of *Oxyrrhis*, more sequence information of taxa is needed, and this is not limited to nuclear rDNA, but also mitochondrial DNA or protein DNA information, which should be added to the concatenated alignment of the phylogenetic tree. However, in the case of mitochondrial DNA or protein DNA, information on various taxa is still lacking, and the degree of differences for each matrix is large, and thus, caution should be taken in the analysis.

Genetic differences between *Oxyrrhis* species can be seen more clearly in the phylogenetic tree based on the nuclear ITS rDNA sequence. According to Lowe et al., the SH and MS strains belonged to *O. marina* clade 1, KRR and GS strains belonged to *O. marina* clade 2, and two Jeju strains belonged to *O. maritima* clade 4 [34]. Lowe et al. conducted a phylogenetic analysis of 5.8S rDNA internal transcribed spacer (5.8S rDNA-ITS) and found two highly divergent lineages within *O. marina* morphospecies, thus proposing the existence of two *Oxyrrhis* spp.: *O. marina* and *O. maritima*. [34]. Based on ITS1-5.8S-ITS2 phylogenetic trees, *O. marina* from SH and MS strains were positioned together with *O. marina* (AY566418, UK) in clade 1, whereas *O. marina* from GS and KRR were positioned with *O. marina* (AY566415, USA) in clade 2. However, *O. maritima* from JJ and HD strains were present on the same branch as that of *O. marina* CCAP 1133/4(AY566417, UK), which showed high divergence from *O. marina* (clades 1 and 2). Moreover, both strains placed highly distinct from *O. maritima* in clade 3 [35]. This study confirmed that the two *Oxyrrhis* species are genetically distinct from each other. Furthermore, we suggest that some strains of *O. marina* (AY566417, UK/ FJ853668, UK/ FJ853677, GRE) mentioned in previous studies should be transferred to *O. maritima*.

The genetic diversity of *O. maritima* has been examined in various studies. Lowe et al. reported that *O. marina* was more widely distributed, from the Atlantic Ocean to the Mediterranean, than *O. maritima* that was partially distributed and elusive in the EU Ocean coasters [35]. Moreover, from previous reports, the genetic divergence in *O. maritima* was limited to clade 3 and clade 4 that contained only 2–3 haplotypes. In this study, however, three clades were discovered throughout the Korean coast. This difference in the dominant biological community might be due to the geographic characteristics of South Korea, which is adjacent to three seas with distinct ecological properties. Therefore, it is worth investigating South Korea for the genetic diversity of *Oxyrrhis* spp. Since *O. marina* is adaptable to various environments, we cannot explain its genetic diversity without analyzing the environmental conditions. Therefore, it is necessary to collect the strains of *O. marina* and *O. maritima* from more diverse environments for analyzing their morphological and genetic characteristics.

*4.3. Physiology*

The environmental factors affecting the growth of *O. marina* and *O. maritima* have been measured in the laboratory. *O. marina* did not grow at temperatures $\geq$35 °C, whereas *O. maritima* showed tolerance to high temperatures, which can explain its habitat. *O. maritima* was collected in tidal waters, which has a significant variation in temperature depending on direct sunlight or weather conditions until high tide.

The effects of salinity on the growth rate examined in this study were also correlated with their habitat. *O. marina* did not grow at a salinity below 4, whereas *O. maritima* species could grow at a salinity of 2. As mentioned above, tidal waters are greatly affected by environmental conditions, and heavy rainfalls can dramatically decrease the salinity close to that of freshwater.

The results indicate that *O. maritima* could survive in a wider range of environmental conditions than *O. marina*, and their ecological characteristics are distinct. In recent years, studies have investigated changes in species and habitats due to an increase in seawater temperature. The comparison of the optimal growth temperature and ecological characteristics of *O. marina* and *O. maritima* in this study can contribute to the accurate distinction between the two species.

Regarding *Oxyrrhis* strains used in the nuclear ITS sequence-based phylogenetic tree, there are no data on their collection areas and ecological characteristics, and thus it is difficult to make a comprehensive judgment on the species classification. As mentioned, the nuclear genome has several limitations such as sequence length. Thus, it should be supplemented by matching genetic information with ecological and morphological information. In this study, by systematically combining morphological information, ecological information, and genetic information, *O. marina* and *O. maritima* were accurately distinguished, and it was confirmed that they show distinct differences in all three aspects.

## 5. Conclusions

*O. marina* is a type species of the genus *Oxyrrhis* and has been used as a model species for a broad range of ecological, biological, and evolutionary studies. However, the taxonomy and classification of *Oxyrrhis* spp. are still vague and controversial, and the other species, *O. maritima*, have not been well investigated to date. For these reasons, we established four different strains of *O. marina* isolated from the coastal waters off GS, MS, SH, and KRR, Korea, and two strains of *O. maritima* isolated from littoral tide pool waters in Jeju Island, Korea, and confirmed that *O. maritima* from JJ and HD were highly divergent from *O. marina* (clades 1 and 2).

Environmental conditions are an important factor in determining the growth of *O. marina* and *O. maritima*. *O. marina* did not grow at water temperatures ≥ 35 °C, whereas *O. maritima* did. Thus, the highest temperature for positive growth of *O. marina* is likely to be approximately 32 °C, whereas that for *O. maritima* is ≥32 °C. *O. marina* did not grow at salinities below 4, and as salinity increased from 5 to 50, the growth rate increased continuously but rapidly decreased when salinity was over 70. However, *O. maritima* could grow at a salinity of 2, and its growth rate increased as salinity increased from 3 to 50, and rapidly decreased as salinity increased over 70. Thus, *O. marina* and *O. maritima* may have different ecophysiology, reflecting their habitats; *O. maritima* inhabits areas of higher salinity and temperatures (tidal pools) than *O. marina* (water columns).

**Supplementary Materials:** The following are available online at https://www.mdpi.com/article/10.3390/w13152057/s1, Table S1: List of primer pairs used for amplification of SSU rDNA segments, ITS region, and LSU rDNA sequences, Table S2: List of species used in constructing nuclear SSU ribosomal (r) DNA phylogenetic trees. Genbank accession numbers are listed to the right of each species, Table S3: List of species used in constructing nuclear ITS1-5.8S-ITS2 phylogenetic trees. Genbank accession numbers are listed to the right of each species.

**Author Contributions:** Data analyses, methodology, writing—original draft preparation, M.K.J.; Culture collection, strain setup, culturing and microscopic analysis, T.Y.Y. and S.J.M.; Conceptualization—review and editing, J.P.; conceptualization, supervision, project administration, writing—review and editing, E.Y.Y. All authors have read and agreed to the published version of the manuscript.

**Funding:** This research was a part of the project titled "Development of eco-friendly functional strategic materials for improving blood circulation derived from marine microalgae (No. 20210515)" and "Improvement of management strategies on marine disturbing and harmful organisms (No. 20190518)" funded by the Ministry of Oceans and Fisheries, Korea.

**Data Availability Statement:** The data presented in this study are available on request from the corresponding authors.

**Acknowledgments:** We thank H. J. Jeong for comments on data analysis, J. H. Hyung and E. J. Kim for technical support and Y. D. Yoo for analysis of ESD.

**Conflicts of Interest:** The authors declare no conflict of interest.

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
