# Peer review of "Taxonomy and Physiology of Oxyrrhis marina and Oxyrrhis maritima in Korean Waters"

_water, doi:10.3390/w13152057_

Round 1
Reviewer 1 Report
The authors present culturing of six Oxyrrhis strains from Korean coastal waters. Each culture was tested for optimal salinity, temperature and light growth conditions. As well rDNA sequences were determined from SSU through the ITS regions and compared with previously described sequences for Oxyrhhis. Because this genus is a cosmopolitan and common heterotrophic member of the dinoflagellate lineages and has not been well described, this contribution will provide an excellent update on this genus in Korean waters. Special attention was paid to integrate previous results (table 4 and figure 7) which are welcome to create context. The paper is mostly well written with a few minor errors and I have several suggestions detailed below.
lines 21 to 24 in the abstract would read better if there were results for temp, salinity and light. In addition, in the results sections strong topic sentences starting the paragraph could be written with a template like this, "The optimal temperature | salinity | light parameters for ___ was (specific or range of values)..."
line 39 -- 'the group' is ambiguous. Specifically, in protein coding gene trees Oxyrrhis is placed intermediate between Perkinsus and the combined clade of syndineans and core dinos.
line 255 delete ' and slightly different ' As this disagrees with the previous phrase.
Reviewer 2 Report
This study entitled “Taxonomy and Physiology of Oxyrrhis marina and Oxyrrhis
maritima in Korean waters” present very interesting and insightful findings of the eco-physiology of both species, with their respective ecological preferences matching their respective habitat conditions. Mostly well written and clear, I really enjoyed reading the detailed description of the ecological part of the study. The phylogenetic part, however, is a lot weaker and need a special attention. Some analyses are mentioned but not presented (ML tree, substitution model test) and others are not introduced but presented (two alignments methods, one not described in the material and method section). Along the same line, the conclusions on phylogenetic relationships are flawed, clade 4 being paraphyletic is not a clade, and O. maritima is not monophyletic. The description of the molecular markers used is not detailed enough and analyses should be improved.
Major recommendations
- Provide a clear description of the molecular markers in the introduction, introduce abbreviations and apply throughout. You should precise the genomic origin of each (mitochondrial vs nuclear?)
- Figure 7 is little informative and flawed (clade 4 is not a clade). I suggest replacing it by the Bayesian tree obtained when combining the two data set, for a global phylogenetic hypothesis based on all available markers.
- Apparently, newly generated sequences were not deposited to Genbank while they should.
Minor recommendations
L14 : « industrial importance » may be add an example to put its importance in perspective?
L19 : « SSU rDNA » mitochondrial? nuclear? Abbreviations should be defined at first introduction as well.
L33: This statement probably holds for HTDs only. May be precise?
L36: “HTDs” singular HTD
L45: replace “lineage” by the corresponding systematic rank.
L59: “rDNA” please define and precise which genome.
L131-132: Please define what SSU is. Is it mitochondrial, nuclear? Does it stand for 12S or else? Likewise, ITS region is located in the nuclear genome. Please precise.
L133: Additional sequences were retrieved from Genbank but what about the newly generated sequences? Did you deposited them in Genbank?
L140-141: It is not clear how the best model was determined for each region. Please precise.
L200: Try replacing “However,…” by “Thus, …”
L191-207: Figure 1 is not cited.
L212: Try replacing “Figure 2” by “Figure 3”.
L220: Try replacing “However,…” by “Thus, …”
L230: Same as above, make clear what each marker acronym stand for, precise the genomic origin (nuclear vs mitochondrial) and apply throughout the manuscript.
L235-236: Material and methods say you used Clustal for aligning sequences…
L237: How were gaps treated: Unknown? Fifth state? Precise.
L239: “Properly aligned…” ? Was it not the case just before?
L258-259: This statement does not belong to result section.
L260: Try replacing “with” by “within”
L265: Clade IV is paraphyletic, thus not a clade and O. maritima is not monophyletic.
L327: Table 4 is more suited to the introduction than discussion section. It provides some crucial information, which put the study in perspective with knowledge gaps. I invite the authors to move it to the introduction and adapt accordingly.
L349: Quite not. This study shows that O. maritima is polyphyletic and encompass probably more than one species, eventually translating into at least 3 Oxyrrhis species in Korean waters.
L357: Clade 4 is not a clade…
L366: Figure 7 => Clade 4 is not monophyletic, thus not a clade. I would like to see a phylogenetic reconstruction based on both alignments analyzed here instead of this Figure 7, which is oversimplified and inaccurate.
L338: Try “the type species of the genus”.
